# Geospatial variation and risk factors for malnutrition among postpartum women in rural Bangladesh

Alexandra L. Bellows[1], Andrew L. Thorne-Lyman[1], Saijuddin Shaikh[2], Hasmot Ali[2], Rezwanul Haque[2], Md. Tanvir Islam[3], Shahnaj Parvin[2], Monica M. Pasqualino[1], Frank C. Curriero[4], Alain B. Labrique[1], Md Iqbal Hossain[5], Amanda C. Palmer[1]*

1 Department of International Health, Johns Hopkins Bloomberg School of Public Health, Baltimore, Maryland, United States of America, 2 JiVitA Maternal and Child Health and Nutrition Research Project, Gaibandha, Bangladesh, 3 BRAC Health Programme, BRAC, Dhaka, Bangladesh, 4 Department of Epidemiology, Johns Hopkins Bloomberg School of Public Health, Baltimore, Maryland, United States of America, 5 Icddr,b, Dhaka, Bangladesh

* acpalmer@jhu.edu

## Abstract

Rising prevalences of overweight and obesity and the continued elevated prevalence of underweight in South Asia have contributed to the double burden of malnutrition. This study evaluates geospatial variation in nutritional status and assesses risk factors for poor nutritional status among postpartum women in rural northwestern Bangladesh. This analysis included postpartum women enrolled in the Protein Plus trial, a cluster-randomized controlled infant feeding trial, from 2018 to 2020. We assessed geospatial variation in nutritional status at the individual level using point pattern analysis and at the spatial area level using global Moran's I. In addition, we ran multivariable multinomial logistic regression models to identify risk factors for being classified as underweight and overweight/obesity in this population. A total of 3,801 women were included in this analysis. The prevalence of underweight and overweight/obesity in this study population was 15.2% and 17.3%, respectively. While no spatial dependence was found at the household level, clustering of underweight and overweight was observed at the community level. In an adjusted analysis, women living in households in the highest socioeconomic (SES) quintile were 51% (Relative risk ratio (RRR): 0.49; 95% CI: 0.34-0.69) less likely to be classified as underweight (p < 0.001) while 2.4 (RRR: 2.37; 95% CI:1.69, 3.32) times more likely to be classified as overweight/obese compared to those in lowest SES quintile (p < 0.001). Higher levels of maternal education and greater food variety were also associated with an increased risk of being classified as overweight/obese. In this area of Bangladesh, we found evidence of the double burden of malnutrition among women 6 months postpartum, with over 30% of postpartum women classified as either overweight or underweight, and nutritional status varied spatially at the community level.

**Data availability statement:** All relevant data are available within the manuscript and its Supporting Information files, except for data containing sensitive or potentially identifiable information. Access to data containing household GPS coordinates is restricted to protect participant confidentiality and privacy.

**Funding:** AB was supported by the Johns Hopkins Procter and Gamble Fellowship and Johns Hopkins Department of International Health Tuition Support. Funding for the original trial was provided by the Bill and Melinda Gates Foundation (OPP1163259). Under the grant conditions of the Bill and Melinda Gates Foundation, a Creative Commons Attribution 4.0 Generic License has already been assigned to the Author Accepted Manuscript version that might arise from this submission. The funders contributed to the trial's design, but played no role in data collection, analysis, or preparation of the manuscript.

**Competing interests:** The authors have declared that no competing interests exist.

## Author summary

Many countries in South Asia are experiencing the double burden of malnutrition with rising prevalences of overweight and obesity and a continued high prevalence of underweight, particularly among women. In this cross-sectional study, we evaluated geospatial variation in nutritional status and assessed risk factors for malnutrition among postpartum women living in a rural region of Bangladesh. Women included in this analysis were enrolled in a cluster-randomized infant feeding trial with anthropometry measured at 6 months postpartum. We found that the prevalence of underweight and overweight/obesity was 15.2% and 17.3%, respectively. Within this rural area, the prevalence of under- and overnutrition varied spatially. In the risk factor analysis, socioeconomic status was found to be protective against the classification of underweight, but women of higher socioeconomic status were at a greater risk of being classified as overweight/obese based on body mass index (BMI). Our findings provide further evidence of the double burden of malnutrition and highlight that even in rural areas, there can be spatial variation in nutritional status. This may be important for nutritional policies and programs to consider to reduce the prevalence of undernutrition while also halting the rise in the prevalence of overweight and obesity.

## Introduction

Worldwide, an estimated two billion adults are classified as overweight or obese, and 462 million adults are classified as underweight [1]. The prevalence of overweight (body mass index (BMI): 25.0-29.9 kg/m$^2$) and obesity (BMI: ≥ 30.0 kg/m$^2$) is rising in all regions of the world [1]. In comparison, the prevalence of underweight (BMI: < 18.5 kg/m$^2$) is still higher than 20% of women in many countries in South Asia [2]. The prevalence of obesity is increasing at a faster rate in low- and middle-income countries (LMICs) compared to high-income countries (HICs), with the greatest increases occurring in rural areas of LMICs [2,3]. Rising prevalences of overweight and obesity and the continued elevated prevalence of underweight in some regions of the world have resulted in communities, households, or individuals experiencing the double burden of malnutrition [2,4].

The consequences of the double burden of malnutrition are both biological and economic [5,6]. Maternal underweight is associated with an increased risk of maternal mortality, preterm birth, and low birth weight [7–11]. Maternal obesity increases a woman's risk for pregnancy complications, including gestational diabetes and pre-eclampsia [7,11–14]. Furthermore, maternal obesity is also associated with a higher likelihood of delayed lactation and increased postpartum weight retention [7,15,16]. In addition, higher BMI is associated with an increased risk of diet-related non-communicable diseases (NCDs), such as diabetes and cardiovascular disease [2,17]. NCDs can reduce an individual's economic earnings, decrease worker productivity, and increase healthcare costs for local and national health systems [18].

Bangladesh is in the earlier stages of the nutrition transition [19,20] and is experiencing rapid economic development and urbanization [21]. Among adult women, modeled data from the 2016 NCD Risk Factor Collaboration indicated a 23.0% prevalence of underweight, while the prevalence of overweight/obesity was 23.2% [2]. In rural areas, women's average BMI is lower than in urban areas in Bangladesh, but age-standardized BMI is increasing at a significantly faster rate in rural areas [3]. Changing food environments, dietary intake, and physical activity patterns are some of the main drivers of changing nutritional status globally [4,19,22–24]. Previous studies have identified household and individual risk factors for nutritional status among women in rural LMIC areas, including household wealth, food security status, educational level, and multiparity [25,26]. Still, few studies have assessed more proximal risk factors of nutritional status, such as dietary intake and food environment. In addition, understanding how nutritional status may vary spatially within a research site and or district may be important for policy and programming initiatives that aim to combat the double burden of malnutrition in communities [27]. This study evaluates geospatial variation in nutritional status and assesses risk factors for malnutrition among postpartum women in northwest Bangladesh.

## Methods

### Ethics statement

Study protocols for the mCARE-II trial were approved by the Johns Hopkins Bloomberg School of Public Health Institutional Review Board, Baltimore, Maryland, and the Bangladesh Medical Research Council, Dhaka, Bangladesh. Study protocols for the Protein Plus trial were approved by the Johns Hopkins Bloomberg School of Public Health Institutional Review Board, Baltimore, Maryland, and the Research and Ethics Review Committees of the International Center for Diarrhoeal Disease Research, Dhaka, Bangladesh.

### Study area

Data for this analysis were collected at the JiVitA Research Site located in the Gaibandha District in Northwestern Bangladesh. This research site was established in 2000 by researchers at Johns Hopkins University [28]. The research site encompasses a rural geographic area of approximately 435km$^2$, with a population of approximately 650,000 individuals.

### Participant selection

This analysis includes women enrolled in the Protein Plus trial, a cluster-randomized controlled infant feeding trial (clinicaltrials.gov: NCT03683667). Details of the Protein Plus trial are described in detail elsewhere [29]. In brief, the Protein Plus trial collected data on a rolling basis from women and infants starting at three months postpartum from 01 September 2018–31 March 2020. Eligible women-infant dyads were identified through a pregnancy surveillance program for the mCARE-II trial, a maternal and infant digital health intervention (clinicaltrials.gov: NCT02909179). Women were considered eligible for pregnancy surveillance if they were married, lived with their husbands, and consented to participate in the surveillance study. Eligible participants were invited to partake in the Protein Plus trial at three months postpartum. Field staff obtained written consent for all participants.

### Data collection

Trained interviewers collected maternal and household demographic information at enrollment for the mCARE-II trial. Surveys collected information on maternal age, maternal education, head of household education, household members' occupations, number of people residing in the household, household assets, and housing characteristics.

For the Protein Plus trial, trained field staff collected data on maternal health, household food security, and dietary intake at three (3), six (6), and twelve (12) months postpartum. Trained fieldworkers collected data on maternal anthropometric measures at three and six months postpartum at the household. Prior to data collection, field workers received

a refresher training on standard anthropometry protocols and were required to achieve a target of 1.0% and 1.5% for inter-observer and intra-observer technical errors of measurement, respectively. Additionally, field workers received regular supervision from the quality control team throughout data collection. At three months postpartum, field interviewers measured women's height three times to the nearest 0.1 cm, and height was averaged across the three measurements. Weight was measured at six months postpartum using a SECA scale with an accuracy of 100g. Weight was measured once and recorded to the nearest gram.

Dietary intake was measured using a seven-day food frequency questionnaire (FFQ) that included 50 food items or food groups. Food items were selected based on formative research of commonly consumed foods in the study population and those that contribute most to micronutrient intake. To measure food security, field staff asked participants a series of nine questions about how often they experienced challenges with food access in the prior six months using a five-point Likert scale (1 = Never, 5 = Most days of the week). Questions on food security were from a validated survey tool developed in the same study area [30].

Geospatial coordinates for households were collected by field staff at the time of demographic household data collection for the mCARE-II trial. In December 2020, field staff conducted a geospatial landmark survey that collected global positioning system (GPS) coordinates of markets and grocery shops within the study area using GPS-enabled tablet devices. A gap between geospatial data collection for households and for landmarks occurred due to restrictions related to the COVID-19 pandemic. We defined markets as a group of five or more permanent shops that were open daily, and grocery shops as standalone vendors that sell groceries (e.g., soap, cigarettes, and packaged foods). Field staff assigned to each sector (an administrative unit of approximately 500 households used for cluster randomization in clinical trials) collected GPS coordinates from all markets and grocery shops within each sector. For data management, we entered all GPS coordinates for households and landmarks into ArcPro (www.ESRI.com) [31].

## Statistical analysis

Analysis was conducted using ArcGIS Pro 2.5 (www.ESRI.com) [31] and The R Statistical Computing Environment [32]. Women were included in this analysis if data were available for height, weight at six months postpartum, and household GPS coordinates. In addition, women included in this analysis were 18 years or older at the time of enrollment. Women were excluded from the analysis if the following covariates were missing: age at enrollment, maternal education, dietary intake, and household food security at six months postpartum. If more than one woman resided within a household, we randomly selected one woman per household to be included in the analysis.

Our primary outcome of interest was nutritional status at six months postpartum. Nutritional status was defined using BMI. We classified women into three categories: underweight (BMI < 18.5 kg/m$^2$), normal weight (BMI: 18.5-24.9 kg/m$^2$) and overweight/obese (BMI ≥ 25.0 kg/m$^2$) [33]. In addition, for a sensitivity analysis, we classified women's nutritional status using Asian-specific BMI-cutoffs: underweight (BMI < 18.5 kg/m$^2$), normal weight (BMI: 18.5-22.9 kg/m$^2$) and overweight/obese (BMI ≥ 23.0 kg/m$^2$) [34]. These lower cutoffs may better reflect cardiometabolic risk in South Asian populations, as the risk of NCDs is higher at lower levels of BMI in Asian populations compared to European populations [33].

First, we assessed geospatial variation in nutritional status (BMI), our main outcome of interest, to better understand the spatial characteristics of the data, to identify high prevalence areas for underweight and overweight within the study area, and as a check on the independence of the data for our subsequent risk factor analysis. We assessed geospatial variation in nutritional status at the household and spatial area levels. At the household level, a semivariogram was calculated to estimate spatial dependence in BMI as a continuous measure and interpreted to assess whether women who live closer to each other have more similar BMI than those who live further away [35]. In addition, we created BMI categories and we conducted a point pattern analysis. We estimated spatial intensity of underweight and overweight (expected number of events per unit area) to map the concentration of cases (underweight/overweight). To characterize smaller-scale spatial variation in the BMI categories, we calculated the difference in K functions to assess whether women who are

categorized as underweight based on BMI cluster more (are more spatially compact) compared to women categorized as normal weight and repeated for the overweight to normal weight comparison [36,37].

At the spatial area level, we assessed spatial dependence using both mauza, an administrative area unit of Bangladesh (n = 146), and research site administrative boundaries (TLPIN) (n = 38). We created choropleth maps for the prevalence of underweight and overweight/obesity within each geographic unit. To assess spatial clustering (how similar is the prevalence of underweight and overweight/obesity for neighboring area units), we calculated global Moran's I, an index of spatial correlation. Neighbors were defined as those that share a boundary (i.e., adjacent). Confidence intervals for global Moran's I were calculated using Monte Carlo methods. In addition, we plotted Moran's I for increasing order of neighbors (lag 1, 2, 3).

Next, we ran bivariate and multivariable multinomial logistic regression models using the "NNET" package in R to identify risk factors for underweight and overweight/obesity. We selected a multinomial model to allow our outcome of interest, nutritional status, to have three categories: underweight, normal weight and overweight/obese. We considered participants categorized as normal-weight to be the reference group. For the multivariable model, we included risk factors with a p-value <0.1 in the bivariate models and a covariate that indicated if weight was measured during Ramadan (yes/no). In the multivariable model, we considered a p-value of <0.05 to be statistically significant. As a sensitivity analysis, we classified women's nutritional status using Asian-specific BMI cutoffs [33] and reran the models described above. We selected household and individual risk factors for multinomial models based on a priori knowledge and those included in risk factor analyses of similar populations [25]. Individual risk factors included maternal age, number of previous live births, maternal education, food variety score (FVS), less healthy food consumption, and season of anthropometric measurement. Household risk factors included a living standard index, food security, market density, and grocery shop density. Definitions of risk factors included in the models are described below.

To quantify dietary intake, we calculated two scores: a continuous FVS to assess consumption of non-starchy staple foods and a dichotomous indicator for consumption of less healthy food items [38]. Diet in Bangladesh can vary seasonally; therefore, to better capture usual intake, we averaged dietary data of non-starchy staple foods from three, six, and 12 months postpartum. Because of the rolling enrollment, dietary data were collected during all seasons. For dietary scores to be calculated in this analysis, a participant must have at least two dietary recalls that did not occur during Ramadan. Dietary recalls during Ramadan were excluded as these are less likely to reflect normal intake [39]. The FVS was constructed by summing the total number of foods/food groups consumed on average at least once in the last seven days. Less healthy food items (defined below) were not included in this score. In the multivariable model, FVS was included as a categorical variable using quintiles based on the distribution of the data. In addition, we calculated a dichotomous variable for above-average less healthy food consumption (>3 times per week), which included the following food items: soda, sweet yogurt, sugar cane, cake, biscuits, mishti (a Bengali sweet), chocolate, candy, ice cream, salty snacks, and foods fried in oil. We considered women who ate these foods three or more times within the last week to be above average consumers (Median: 2.67). We defined season of anthropometric measurement as winter (mid-December to mid-February), spring (mid-February to mid-April), summer (mid-April to mid-June), early monsoon (mid-June to mid-August), late monsoon (mid-August to mid-October), and autumn (mid-October to mid-December) [40].

Household wealth was measured by a living standard index (LSI). This index was created using principal component analysis of household assets and housing characteristics, with the first principal component used to capture variance in wealth [24,25]. In all models, we included LSI as a categorical variable, with the index divided into quintiles based on the distribution of that data. For household food insecurity scores, we summed the scores of the nine questions regarding food access and categorized households into three main categories: food secure (9 points), mild to moderate food insecurity (10–15 points), and severe food insecurity (>15 points) [30].

We used density of markets and grocery shops around a woman's home as proxy measures for a woman's external food environment [41]. We selected density measures as a proxy of food environment because a previous analysis found a

relationship between density of markets and dietary intake [38]. In addition, other studies have found an association between the density of food vendors and nutritional status [42,43]. To measure the density of food vendors, we imported geospatial data into ArcPro and projected coordinates to EPSG: 3106 (Gulshan 303 Bangladesh Transverse Mercator) to calculate buffer areas around each household in meters. For each household, we defined the density of markets as the number of markets within 1600m of a household and the density of grocery shops as the number of grocery shops within 400m of a household. We chose these boundaries to reflect density at the village level for markets and immediate proximity for grocery shops [44,45]. For this analysis, we categorized the density of markets and grocery shops using quartiles based on the distribution of data.

## Results

A total of 3,801 women were included in this analysis. In the Protein Plus trial, 5,891 mother-infant dyads were enrolled. For this analysis, we excluded 834 (14.2%) women who were missing anthropometric measures at six months postpartum, 901 (15.3%) women who were less than 18 years of age or missing age at enrollment, and 229 (3.9%) women from households with missing geospatial coordinates (food environment assessment). We excluded 3 women with implausible BMI ($<12$ kg/m$^2$) and 19 (0.3%) observations with greater than +/- 10 kg weight change from 3 months to 6 months postpartum. An additional 85 (1.4%) women were excluded for missing risk factors of interest, and 19 (0.3%) women were removed after randomly selecting one woman per household **(Fig A in** S1 Text). Demographic characteristics of women included and excluded in this analysis are displayed in **Table A in** S1 Text.

Of the 3,801 women included in the analysis, 579 (15.2%) and 658 (17.3%) were classified as underweight and overweight/obese based on BMI, respectively, at six months postpartum (Table 1). Maternal short stature (height <145 cm) was most prevalent among women classified as underweight (20%) compared to women classified as normal weight (18%) or overweight (13%). Average age of women included in the study was approximately 25 years (SD: 4.9) and the majority of women (68%) obtained some primary education. Approximately 30% of households reported mild to moderate food insecurity, and 68% of women reported no paid employment (Table 2).

When assessing geospatial variation at the household level, we found no evidence of spatial dependence for BMI either as a continuous measure (BMI semivariogram was relatively flat) or categorical (difference in k functions were within confidence bands) **(Figs B-D in** S1 Text). Although the spatial concentration of women classified as underweight revealed an east west gradient, this was not consistent with spatial concentration in overweight cases (**Fig E & F** in S1 Text). The prevalence of underweight and overweight showed geospatial dependence when aggregated to the research site administrative units (TLPIN) (Fig 1) and mauza (**Fig G in** S1 Text), indicating spatial variation at the community level. Moran's I value was 0.38 (p-value = 0.001) for the prevalence of underweight (**Fig H in** S1 Text) and 0.29 (p-value = 0.001) for the prevalence of overweight (**Fig I in** S1 Text), indicating similarity among neighboring research site administrative units. This indicates that for both the prevalence of underweight and overweight, there is greater clustering of areas with high or low prevalences than would be expected if the distribution of area-level nutritional status were spatially random.

**Table 1. Descriptive statistics of anthropometric measures and indices for women enrolled among postpartum women in rural Bangladesh (n = 3,801).**

| | All participants | Underweight | Normal Weight | Overweight |
|---|---|---|---|---|
| | | (BMI < 18.5 kg/m$^2$) | (BMI 18.5-24.9 kg/m$^2$) | (BMI ≥ 25.0 kg/m$^2$) |
| N | 3,801 | 579 (15.2) | 2564 (67.5) | 658 (17.3) |
| Height (cm) | 149.91 ± 5.25 | 149.68 ± 5.34 | 149.79 ± 5.27 | 150.57 ± 5.01 |
| Maternal short stature (<145 cm) | 666 (17.5) | 117 (20.2) | 462 (18.0) | 87 (13.2) |
| Weight at 6 months postpartum (kg) | 49.14 ± 8.56 | 38.97 ± 3.41 | 48.14 ± 5.40 | 62.01 ± 6.24 |
| BMI at 6 months postpartum (kg/m$^2$) | 21.83 ± 3.37 | 17.37 ± 0.90 | 21.42 ± 1.77 | 27.32 ± 2.07 |

*Values presented are mean ± SD or n (%)*

**Global Public Health**

PLOS

**Table 2. Baseline characteristics of postpartum women and households by nutritional status in rural Bangladesh (n = 3,801).**

| | All Participants | Underweight | Normal Weight | Overweight |
|---|---|---|---|---|
| | | (BMI < 18.5 kg/m²) | (BMI 18.5-24.9 kg/m²) | (BMI ≥ 25.0 kg/m²) |
| N | 3,801 | 579 (15.2) | 2564 (67.5) | 658 (17.3) |
| **Individual Factors** | | | | |
| Age | 24.74 ± 4.89 | 23.89 ± 4.83 | 24.63 ± 4.83 | 25.90 ± 4.98 |
| Number of previous live births | 1.67 ± 0.90 | 1.65 ± 0.92 | 1.67 ± 0.90 | 1.67 ± 0.87 |
| **Maternal education** | | | | |
| No Schooling | 500 (13.2) | 81 (14.0) | 350 (13.7) | 69 (10.5) |
| Class 1–9 | 2589 (68.1) | 411 (71.0) | 1775 (69.2) | 403 (61.2) |
| SSC Passed | 228 (6.0) | 26 (4.5) | 146 (5.7) | 56 (8.5) |
| 11 years + | 484 (12.7) | 61 (10.5) | 293 (11.4) | 130 (19.8) |
| **Maternal occupation** | | | | |
| No paid work | 2379 (62.6) | 366 (63.2) | 1622 (63.3) | 391 (59.4) |
| Own business | 1159 (30.5) | 164 (28.3) | 782 (30.5) | 213 (32.4) |
| Laborer | 96 (2.5) | 27 (4.7) | 59 (2.3) | 10 (1.5) |
| Private service | 118 (3.1) | 14 (2.4) | 76 (3.0) | 28 (4.3) |
| Farmer/sharecropper | 27 (0.7) | 8 (1.4) | 14 (0.5) | 5 (0.8) |
| Other | 22 (0.6) | 0 (0.0) | 11 (0.4) | 11 (1.7) |
| Food variety score[a] | 9.80 ± 3.64 | 9.21 ± 3.31 | 9.66 ± 3.58 | 10.84 ± 3.93 |
| Less healthy food consumption (times/week)[b] | 3.42 ± 3.21 | 2.97 ± 2.94 | 3.36 ± 3.18 | 4.07 ± 3.45 |
| **Household Factors** | | | | |
| Market density (1600m) | 5.03 ± 3.02 | 4.70 ± 2.64 | 5.05 ± 2.99 | 5.24 ± 3.41 |
| Variety store density (400m) | 4.62 ± 3.17 | 4.53 ± 3.11 | 4.66 ± 3.22 | 4.51 ± 3.03 |
| Household size | 4.40 ± 1.86 | 4.37 ± 1.85 | 4.34 ± 1.77 | 4.67 ± 2.17 |
| **Wealth quintile[c]** | | | | |
| 1 (lowest) | 761 (20.0) | 168 (29.0) | 515 (20.1) | 78 (11.9) |
| 2 | 760 (20.0) | 127 (22.9) | 542 (21.2) | 91 (13.8) |
| 3 | 760 (20.0) | 97 (16.8) | 530 (20.7) | 133 (20.2) |
| 4 | 760 (20.0) | 106 (18.3) | 511 (19.9) | 143 (21.7) |
| 5 (highest) | 760 (20.0) | 81 (14.0) | 466 (18.2) | 213 (32.4) |
| **Food security[d]** | | | | |
| Food secure | 2510 (66.0) | 384 (66.3) | 1671 (65.2) | 455 (69.1) |
| Mild to moderate food insecurity | 1180 (31.1) | 175 (30.1) | 812 (31.7) | 193 (29.3) |
| Severe food insecurity | 111 (2.9) | 20 (3.5) | 81 (3.1) | 10 (1.5) |

SSC: Secondary School Certificate

Values presented are mean ± SD or n (%).

[a]Food variety scores are defined as the average number of non-starchy staple foods items or groups consumed in the last week excluding sweet & salty snacks and sugary sweetened beverages. Scores range from 1-25.

[b]Less healthy food consumption is defined as consumption of foods such as soda, salty snacks, sugar cane, etc. at least 3 times on average in the 7-day recall period.

[c]Wealth was characterized by a living standards index (LSI) and calculated using principal component analysis (PCA) from household assets and housing characteristics.

[d]Household food insecurity scores were calculated by summing responses from 9 food behavior questions and categorized as follows: food secure (9 points), mild to moderate food insecurity (10–15 points), and severe food insecurity (>15 points) [30]. Participants were asked about frequency of behavior in the last 6 months using a 5-point Likert scale (1 = Never, 5 = Most days of the week).

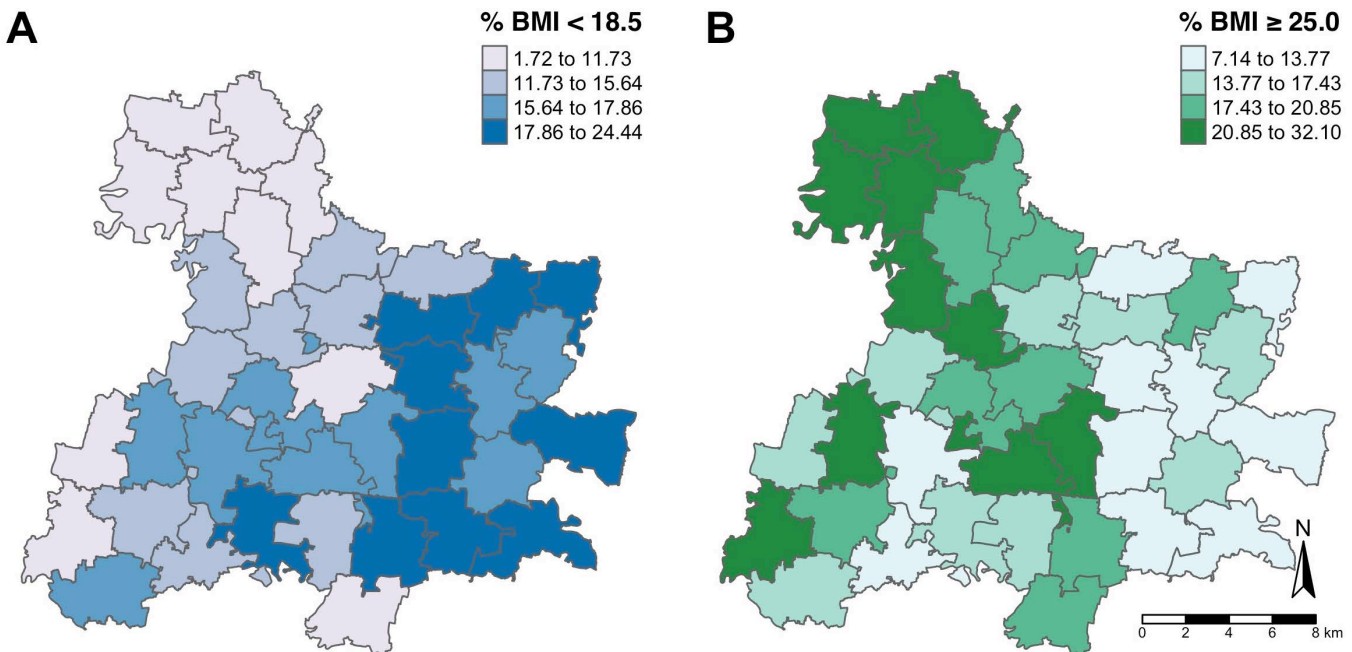

**Fig 1. Prevalence of A) underweight (BMI < 18.5 kg/m²) and B) overweight/obesity (BMI ≥ 25.0 kg/m²) at six months postpartum by TLPIN[a] among women enrolled in the Protein Plus trial (n = 3,801).** *Cutoffs for each map are based on quartiles of prevalence A) underweight (BMI < 18.5 kg/m²) and B) overweight/obesity (BMI ≥ 25.0 kg/m²).* [a] TLPIN is an administrative unit used for research study purposes.

Bivariate multinomial logistic regression models indicated that younger maternal age, lower market density, lower food variety scores, below-average less healthy food consumption, and lower household SES were associated with an increased risk of being classified as underweight (p < 0.1). In addition, older maternal age, higher maternal education, higher FVS, above-average less healthy food consumption, higher household SES, and household food security were associated with an increased risk of being classified as overweight/obese (p < 0.1). Maternal short stature was also associated with a reduced risk of being classified as overweight/obese (p < 0.01) **(Table B in** S1 Text**)**.

Multivariable multinomial logistic regression models indicated that maternal age and household SES were inversely associated with the risk of being classified as underweight and positively associated with the risk of being classified as overweight/obese. Postpartum women living in households in the highest SES quintile were 51% (RRR: 0.49; 95% CI: 0.34-0.69) less likely to be classified as underweight (p < 0.001) while 2.4 (RRR: 2.37; 95% CI:1.69, 3.32) times more likely to be classified as overweight/obese compared to women living in households in lowest SES quintile (p < 0.001) (Fig 2 **&  Table C in** S1 Text)

After controlling for other risk factors, above-average consumption of less healthy foods and living in households with the highest market density remained inversely associated with the risk of being classified as underweight. Postpartum women who consumed less healthy food options >3 times per week had an 18% (RRR: 0.82; 95% CI: 0.67-1.00) decreased risk of being classified as underweight (p = 0.05). Compared to postpartum women living in households with ≤2 markets within 1600m of their household, women who lived in households with >6 markets within 1600m had a 28% (RRR: 0.72; 95% CI: 0.55-0.95) reduction in risk of being classified as underweight (p = 0.02) (Fig 2 **& Table C in** S1 Text).

After controlling for other risk factors, women who attained the highest levels of education (grade 11 or above) had a 1.5 (RRR: 1.53; 95% CI: 1.03, 2.27) times greater risk of being classified as overweight or obese (p = 0.04) compared to women who received no formal schooling. In addition, women in the top quartile for FVS were 1.4 (RRR: 1.38; 95% CI:

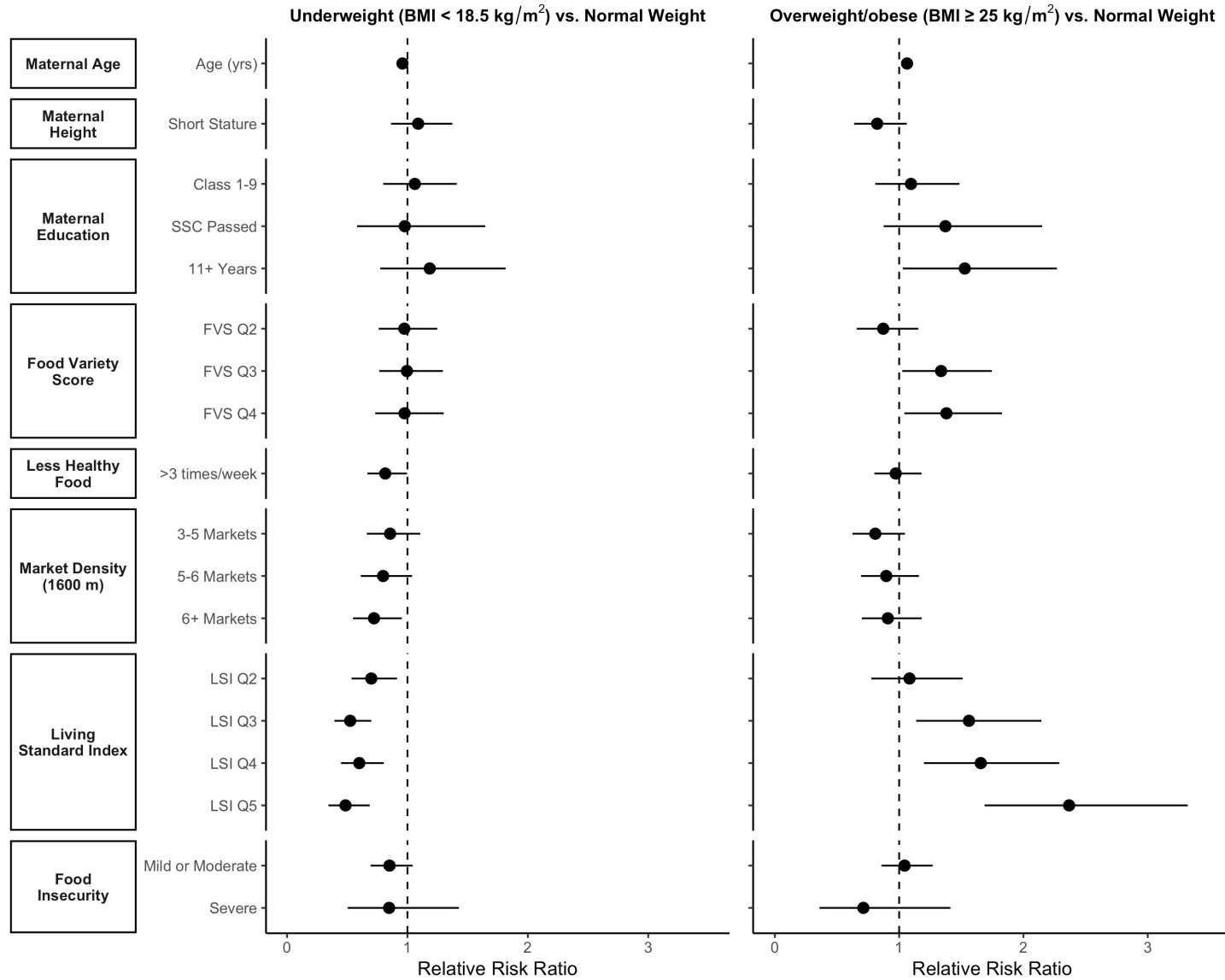

**Fig 2. Relative risk ratios and 95% confidence intervals for the association between nutritional status and selected risk factors from multivariate multinomial logistic regression models among postpartum women in rural Bangladesh (n = 3,801).** Models included the following covariates: maternal age (continuous), maternal short stature (categorical, reference = ≥ 145 cm), maternal education (categorical, reference = "No schooling"), food variety scores (categorical, reference = lowest quartile), less healthy food consumption (categorical, reference = > 3 times/week), market density (categorical, reference = 0–2 markets), living standard index (categorical, reference = lowest quartile), food insecurity (categorical, reference = food secure) and 6 month maternal survey conducted during Ramadan (binary, reference = no). SSC: Secondary school certificate, FVS: food variety score, LSI: Living standard index Short stature is defined as maternal height < 145 cm. Food variety scores are defined as the average number of non-starchy staple food items or groups consumed in the last week, excluding sweet & salty snacks and sugary sweetened beverages. Scores range from 1-25. Less healthy food consumption is defined as consumption of foods such as soda, salty snacks, sugar cane, etc., at least 3 times on average in the 7-day recall period.

1.04, 1.83) times more likely to be classified as overweight or obese compared to women consuming the lowest quartile (p < 0.02). Short stature, less healthy food consumption, market density, and food insecurity were not associated with the risk of being classified as overweight/obese after controlling for other risk factors (Fig 2 & **Table C** in S1 Text). The multivariable model also did not exhibit residual spatial variation (spatial dependence in the regression residuals), thus

satisfying an assumption of the model (**Fig J** in S1 Text). In addition, similar results to the bivariate and multivariate models were found in sensitivity analyses using Asian-specific BMI categories (**Tables D & E** in S1 Text).

## Discussion

Over 30% of postpartum women included in our analysis were classified as either under (15%) or overweight (17%), indicating the presence of the double burden of malnutrition within this community. We did not find evidence of spatial dependence at the household level, but when aggregated to the community level, we found clustering of high and low prevalences of both underweight and overweight. Postpartum women who were from higher SES households, older in age, lived in households with higher market density, and consumed less healthy food options were at decreased risk of being classified as underweight. On the other hand, women who lived in higher SES households, ate a more diverse diet, and obtained a secondary education were at greater risk of being classified as overweight or obese.

Although our results are from an area of Bangladesh with many nutrition-related interventions and studies [28,46,47], the prevalences of undernutrition and overweight/obesity in this study population are similar to other studies from rural Bangladesh, indicating that results from our risk factor analysis may be generalizable to similar populations in rural areas in Bangladesh. The average BMI in our study population was 21.8 kg/m$^2$, which is consistent with modeled estimates of BMI in rural Bangladesh from 2017 [3]. On the other hand, in comparison with the 2017–2018 Demographic and Health Survey (DHS), we found a higher prevalence of underweight and a lower prevalence of overweight and obesity among ever-married women in rural areas. Estimates from the DHS indicate that 13% of ever-married women were classified as underweight based on BMI, while 28% of ever-married women were classified as overweight or obese based on BMI in rural areas [48]. Another study that explored risk factors of nutritional status using anthropometric data from 2011 in the Sylhet Division of Bangladesh estimated a significantly higher percentage of underweight women (37.0%) and a lower prevalence of overweight and obesity (7.2%) compared to our study [25]. This is consistent with findings from DHS, which indicate that the Sylhet Division has the highest prevalence of undernutrition for ever-married women [48]. Differences may also be attributable to temporal trends in BMI between studies. Studies that have looked at the change in BMI in Bangladesh have shown that the average BMI is rising most rapidly in rural areas of the country [3,49,50].

Within the JiVitA study site, we found geospatial variation in nutritional status at the area level; however, at the household level, we found no evidence of geospatial variation. The lack of statistically significant spatial dependence at the household level is most likely due to high population density and variability in nutritional status among neighboring women within the same area unit. At the area level, we see a higher prevalence of undernutrition in the eastern quadrant of the research site and a higher prevalence of overweight and obesity in the northern and western quadrants. Interestingly, few mauzas have a high prevalence of both under- and overnutrition.

Characterizing geospatial variation in nutritional status is important for identifying potential high-priority areas for program and policy interventions. Other studies have examined the geospatial variation of maternal nutritional status within larger geographic regions, such as country- or state-level [27,51]. For example, in Bihar, India, a study found that trends in several reproductive, maternal and child health indicators varied spatially at the block-level during a state-wide scale-up of a maternal and child health intervention [27]. Moreover, a nationally representative analysis in Afghanistan found geospatial variation at the province level with higher prevalences of underweight for women of reproductive age in the north and northeast provinces [51]. Additionally, more research has been conducted on the geospatial variation in the nutritional status of children compared to adults. Researchers with the Local Burden of Disease Project have mapped the prevalence of stunting (height for age<-2 SD), wasting (weight for height<-2 SD), and overweight (BMI-for age>2 SD) in children under five years of age at the 5 km x 5 km geospatial unit [52,53]. Researchers found significant geospatial variation in child nutritional status within aggregated administrative units [52,53]. For example, in India, spatial analysis was conducted to assess differences in child growth factors, which found differences in trends between districts within the same state [54].

Programs seeking to improve the nutritional status of the population may want to tailor their approaches depending on the community-level prevalence of undernutrition and overnutrition.

We identified individual and household risk factors that were inversely associated with the risk of being classified as underweight at 6 months postpartum. These included maternal age, household SES, above-average consumption of less healthy food options, and increased market density. The relationship between both maternal age and household SES with undernutrition in women of reproductive age has been well documented in Bangladesh and other LMIC contexts [25,55–58]. Surprisingly, food security status was not associated with the risk of being classified as underweight in either unadjusted or adjusted models. Similarly, higher FVS was inversely associated with the risk of undernutrition in our unadjusted analysis, but once we added other risk factors to the model, this association was no longer significant. This is likely due to collinearity with household SES, as households with higher SES status are also more likely to eat more diverse diets and be more food secure.

Overweight and obesity were more prevalent among older women, those with a more diverse diet, and those in households with a higher SES. Again, results for maternal age and household SES were consistent with findings from other studies [25,50,56,57]. Of interest is the positive association between FVS and the risk of being classified as overweight/ obese, even after controlling for household wealth. A similar association was reported from a study in Sri Lanka [59] and among adolescents in Bangladesh [60]. Increasing women's dietary diversity is the focus of many programs and initiatives in rural LMIC settings [61–64]. Dietary diversity scores were originally developed as a proxy measure for micronutrient adequacy [62,65]. They were not developed to include aspects of moderation, which is an important component as dietary patterns change with the ongoing nutrition transition in LMICs. In settings where the double burden of malnutrition is rising, it will be important for programs and policy initiatives aimed at improving diet quality to measure both diversity and moderation components of the diet. We were surprised to find no association between less healthy food options and the risk of being classified as overweight/obese. This is potentially due to the limited consumption of these foods among women included in our study. Studies from other settings have found strong associations between less healthy food consumption and risk of overweight and obesity, but the amount of less healthy food options was higher [66–69]. Ultra-processed foods are becoming increasingly more available and affordable in LMICs [70,71]. While limited data currently exist for processed food consumption in Bangladesh, modeled data from Euromonitor suggest a 67% increase in ultra-processed foods sales from 2013 to 2018 [72]. Additionally, another analysis found that the presence of small grocery shops that typically sell packaged/processed foods has increased significantly in this study area from 2004 to 2020 [73]. As access to less healthy food options increases, it is likely we will begin seeing a greater association between less healthy food consumption and overweight/obesity in this setting.

A novel feature of this analysis is the exploration of spatial dependency of nutritional status within the study area. While we did not find statistically significant spatial dependence at the household level, our geospatial analysis allowed us to confirm independence of outcomes for our multinomial logistic regression models, a key regression assumption. In addition, our analysis was able to include proximal risk factors of nutritional status, such as food environment and dietary intake measures, which are often understudied in rural LMIC contexts. This study does have some limitations, including the lack of anthropometric measures pre-pregnancy. Since BMI is likely to fluctuate post-pregnancy, we are at risk of potential misclassification of non-pregnancy-related nutritional status, which may result in an underestimation of underweight women and an overestimation of overweight/obese women. BMI is a controversial proxy measure for adiposity. At the population level, the correlation between BMI and adiposity is relatively strong, but there are differences by age, race, and gender [74]. For example, there has been much debate over appropriate BMI cutoffs for classifications of overweight/obesity in Asian populations, as evidence suggests people of Asian descent are at higher risk of developing type-II diabetes and cardiovascular disease at lower BMI cutoffs [33,75]. Previous research from the JiVitA study team found that adult women living in the study area had a higher percentage of body fat at lower BMI cutoffs than other populations [76]. Therefore, we included a sensitivity analysis using Asian-specific cutoffs for BMI. Moreover, the dietary tool we used

to assess diet quality did not measure quantity and was restricted to a set number of foods. Therefore, the diet quality indicators included in the models did not take into account how much of each food item women were eating. Finally, confounders included in our multinomial model were limited to the data available; therefore, our models may be missing key risk factors and susceptible to residual confounding. As this is a cross-sectional analysis, we are only able to assess association rather than causation.

In the last decade, the government of Bangladesh has launched additional initiatives to improve the nutritional status of its population. In 2013, the government drafted the first set of quantitative dietary guidelines for Bangladesh [77], and in 2016, the Ministry of Food in Bangladesh released a four-year strategic plan prioritizing nutrition-sensitive food systems initiatives [78]. The four-year strategic plan aims to improve the food system to make it more sustainable and ensure that everyone has access to and can afford a healthy diet. Priority interventions include: improving market infrastructures, promoting public-private partnerships to reduce all forms of malnutrition, promoting healthy diets to prevent and control NCDs, and promoting nutrient-dense foods to prevent undernutrition. Results from this analysis highlight the need to prioritize "double duty" programs and policies that prevent all forms of malnutrition, ensuring policies or programs focused on one form of malnutrition also minimize the risk of other forms [79,80]. Policy and programs should also be aware of potential geospatial variation in nutritional status, even in predominantly rural areas, and tailor interventions based on the prevalence of nutritional status within communities.

## Supporting information

**S1 Text. Supplemental Figures and Tables.**
(DOCX)

**S1 Data. Anonymised analytical dataset.**
(XLSX)

## Acknowledgments

We express our gratitude to the community leaders in the Gaibandha District and the women and their families who participated in the JiVitA-6 trial. We would also like to thank Dr. Yeeli Mui for her mentorship and initial comments on the analysis.

## Author contributions

**Conceptualization:** Alexandra L Bellows, Amanda C. Palmer.

**Data curation:** Saijuddin Shaikh, Hasmot Ali, Rezwanul Haque, Md. Tanvir Islam, Shahnaj Parvin, Monica M Pasqualino, Alain B. Labrique.

**Formal analysis:** Alexandra L Bellows, Frank C Curriero.

**Funding acquisition:** Alain B. Labrique, Md Iqbal Hossain, Amanda C. Palmer.

**Methodology:** Andrew L. Thorne-Lyman, Rezwanul Haque, Monica M Pasqualino, Frank C Curriero, Md Iqbal Hossain, Amanda C. Palmer.

**Project administration:** Saijuddin Shaikh, Hasmot Ali, Rezwanul Haque, Md. Tanvir Islam, Shahnaj Parvin, Alain B. Labrique.

**Supervision:** Andrew L. Thorne-Lyman, Frank C Curriero, Amanda C. Palmer.

**Visualization:** Alexandra L Bellows, Frank C Curriero.

**Writing – original draft:** Alexandra L Bellows.

**Writing – review & editing:** Andrew L. Thorne-Lyman, Saijuddin Shaikh, Hasmot Ali, Rezwanul Haque, Md. Tanvir Islam, Shahnaj Parvin, Monica M Pasqualino, Frank C Curriero, Md Iqbal Hossain, Amanda C. Palmer.

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
