## [Decision Letter · Decision Letter 0]

19 Mar 2025

PGPH-D-24-03004

Geospatial variation and risk factors for malnutrition among postpartum women in rural Bangladesh

Dear Dr. Bellows,

Thank you for submitting your manuscript to PLOS Global Public Health. After careful consideration, we feel that it has merit but does not fully meet PLOS Global Public Health’s publication criteria as it currently stands. Therefore, we invite you to submit a revised version of the manuscript that addresses the points raised during the review process.

Two reviewers have provided their detailed comments below. Please respond to each comment in turn and make the necessary amendments in your manuscript. I would like to bring your attention in particular to comments surrounding the reporting of your methodology. Please note it is a requirement of publication in PLOS Global Public Health that experiments, statistics, and other analyses are performed to a high technical standard and are described in sufficient detail (https://journals.plos.org/globalpublichealth/s/criteria-for-publication#loc-3). If this criterion is not met we may have to issue a reject decision.

We look forward to receiving your revised manuscript.

Kind regards,

Joanna Tindall, PhD

Staff Editor

Journal Requirements:

Additional Editor Comments (if provided):

Reviewers' comments:

Reviewer's Responses to Questions

**Comments to the Author**

1. Does this manuscript meet PLOS Global Public Health’s publication criteria?

Reviewer #1: No

Reviewer #2: Yes

2. Has the statistical analysis been performed appropriately and rigorously?

Reviewer #1: No

Reviewer #2: Yes

3. Have the authors made all data underlying the findings in their manuscript fully available (please refer to the Data Availability Statement at the start of the manuscript PDF file)?

Reviewer #1: Yes

Reviewer #2: Yes

4. Is the manuscript presented in an intelligible fashion and written in standard English?

Reviewer #1: Yes

Reviewer #2: Yes

Reviewer #1: abstract

1. Enhance the Spatial Analysis Component

Clearly define how geospatial variation was measured and analyzed.

Provide details on the spatial clustering methodology (e.g., Moran’s I, spatial regression models).

Explain why no spatial dependence was found at the household level but observed at the community level.

2. Clarify the Methodological Approach

Justify the use of multinomial logistic regression and discuss its limitations.

Address potential biases, including sample selection and confounding variables.

Specify whether the Protein Plus trial design influenced the study's generalizability.

3. Improve Contextualization and Interpretation

Compare findings with other studies on malnutrition in similar rural South Asian settings.

Discuss broader implications, particularly how these findings can inform public health policies.

Explain the mechanisms by which education and socioeconomic status influence nutritional status.

4. Strengthen the Conclusion

Move beyond simply stating the double burden of malnutrition—propose potential interventions.

Discuss how policymakers or healthcare providers can address these disparities.

Acknowledge study limitations (e.g., self-reported dietary intake, potential measurement errors).

1. Improve Clarity and Structure

The section contains dense information that could be better structured for readability. Breaking it into subsections (e.g., study site, participant selection, ethical considerations) would enhance clarity.

Define JiVitA Research Site at the beginning instead of mentioning it mid-sentence.

2. Justify Study Site Selection

The authors should explain why Gaibandha district was chosen for the study.

Provide context on the malnutrition burden in this area compared to other regions in Bangladesh.

3. Address Selection Bias

The eligibility criteria (women must be married and live with husbands) exclude single mothers or women in other family structures, which could introduce bias.

Discuss how this exclusion might affect the generalizability of findings to all postpartum women in rural Bangladesh.

4. Ethical Considerations

While approvals are mentioned, it's unclear whether informed consent was obtained from participants for this specific analysis, not just for the primary trials (Protein Plus and mCARE-II).

Did women have the option to withdraw from the study after enrollment in pregnancy surveillance?

5. Clarify Data Collection Timeline

The study mentions data collection at three months postpartum, but it’s unclear if all women were enrolled at the same time or if enrollment was rolling.

Specify if seasonality in food availability or health conditions could have influenced nutritional outcomes.

1. Improve Consistency and Completeness in Anthropometric Measurements

Height was measured three times, while weight was measured only once—why the inconsistency?

Weight should also be measured multiple times to reduce measurement error.

Clarify why height was measured at three months but weight at six months postpartum—this gap could affect BMI calculations.

2. Address Potential Biases in Dietary Intake Assessment

The seven-day food frequency questionnaire (FFQ) is prone to recall bias—consider discussing its limitations.

The FFQ included 50 food items—did it capture seasonal variations in diet?

Explain whether FFQ responses were validated against any other dietary intake methods (e.g., 24-hour recall).

3. Clarify Food Security Measurement

The five-point Likert scale for food security assessment lacks a clear interpretation—how were responses categorized (e.g., food secure vs. insecure)?

Explain whether the nine-question food security survey aligns with any established food security measurement tools (e.g., Household Food Insecurity Access Scale).

4. Improve Clarity on Geospatial Data Collection

The study collected household GPS coordinates in 2018–2020 but conducted a geospatial landmark survey in December 2020—why the time gap?

Define how proximity to markets/variety stores was analyzed—was distance used as a continuous variable, or were categorical thresholds applied?

Consider potential errors in GPS data collection, especially in rural areas where accuracy may be lower.

5. Justify Market and Variety Store Definitions

The definition of markets (≥5 permanent shops open daily) and variety stores (standalone vendors selling groceries) seems arbitrary.

Explain how these definitions impact findings—did proximity to these markets correlate with nutritional outcomes?

Overall Suggestions

Ensure consistency in anthropometric data collection to minimize errors.

Acknowledge recall bias in dietary assessment and justify the FFQ approach.

Provide more detail on geospatial analysis, including how spatial relationships were assessed.

Clarify and justify definitions of food security and market access to ensure meaningful interpretations.

The result tables are not understandable.

Reviewer #2: Overall, the manuscript provides an interesting exploration of geospatial variation in risk factors that adds to the literature on population nutrition.

General comment (abstract conclusions, discussion, etc): The study population is post-partum women. Given that you are unable to adjust for pre-pregnancy weight or weight retention, suggest caution in generalizing conclusions beyond this population. Suggest that sentences summarizing the evidence – including the conclusion statement of the abstract – clearly state that findings are among post-partum women.

Abstract: Please add additional details on analytic methods for assessing geospatial variation to the abstract (methods section).

Methods: Were all women who were enrolled in the Protein Plus trial eligible for this analysis regardless of birth outcome?

Methods: The parent study supported postnatal care (including reminders). Did you explore/account for difference across study arms?

Methods: From the dataset it appears that weight at 3 months post-partum was available. Was weight at enrollment in the mCARE-II trial also available? Is there a reason that these longitudinal weight measurements were not included in the models?

Methods: Anthropometric data quality can impact evaluation of nutritional status. Often different enumerators are assigned to different geographic areas of the study catchment area in a way that could impact analysis of geospatial clustering. Can you describe what steps you took to explore anthropometric data quality and rule out the possibility that different quality achieved by enumerators confounded the results.

Results: You report that 14.2% of the sample were missing weight which is a reasonably high proportion. Supplemental figures – helpfully - describe demographic characteristics of the sample of individuals included v. excluded from analysis (primarily due to missing weight). However, can you speak to whether there was an geospatial clustering of individuals excluded from analysis?

Results: The authors explore the relationship between individual risk factors and community risk factors with nutritional status. However, it seems that there is a unique opportunity to explore the extent individual covariates of interest geographically cluster in a way that impacts nutritional status. For example, you find that LSI is associated with nutritional status but is living in a poorer neighborhood (cluster median LSI) independently associated with nutritional status? What about neighborhoods that are on average younger and/or less educated?

Results: Paragraph beginning on line 282 includes a lot of double negative phrasing. Consider revising.

Results: Supplementary Figures 2 and 10 - Semivariogram for Maternal BMI and associated residuals appear identical (except for the title). Can you confirm that the correct figures are included.

Results/Dataset. Assuming that m3weight and m6 weight refer to weight at 3 and 6 months post partum respectively, there are many women with implausible changes in weight (and therefore BMI). To use an extreme example, there are women with 32kg losses and 38kg gains over a 3-month time period. Over 1% of the sample having +/-6kg weight change over 3 months. Please discuss what data quality reviews (particularly for the anthropometric data) were performed. How would exclusion of implausible values impacts findings?

Discussion: The lack of spatial variation at the household level given clear variability in community-prevalence is the key and somewhat surprising finding. To what extent does this have to do with parameterization (e.g., the use of the continuous v. categorial variable). Do you see the same effect if you look at mean BMI for an area rather than prevalence? Do we see the same phenomenon in previous literature? To give confidence that these findings are not spurious consider: (1) adding figures similar to supplemental Figures 6 and 7 but with mean BMI, (2) including additional documentation in the results on model fit / validation.

**Do you want your identity to be public for this peer review?** For information about this choice, including consent withdrawal, please see our Privacy Policy

Reviewer #1: No

Reviewer #2: No

---

## [Decision Letter · Decision Letter 1]

29 Aug 2025

PGPH-D-24-03004R1

Geospatial variation and risk factors for malnutrition among postpartum women in rural Bangladesh

Dear Dr. Bellows,

Thank you for submitting your manuscript to PLOS Global Public Health. After careful consideration, we feel that it has merit but does not fully meet PLOS Global Public Health’s publication criteria as it currently stands. Therefore, we invite you to submit a revised version of the manuscript that addresses the points raised during the review process.

Your manuscript has been assessed by one new reviewer; the previous reviewers were not available. The new reviewer's comments are available below.

The new reviewer has indicated that they are satisfied with your revisions in response to the previous reviewers' comments, and has requested some additional clarifications that require addressing before publication. These relate to the discussion of other proximate determinants, as well as justification for some aspects of the methodology. Please ensure you address each of the reviewers' comments when revising your manuscript.

We look forward to receiving your revised manuscript.

Kind regards,

Hugh Cowley

Staff Editor

Journal Requirements:

Additional Editor Comments (if provided):

Reviewers' comments:

Reviewer's Responses to Questions

**Comments to the Author**

Reviewer #3: All comments have been addressed

publication criteria?

Reviewer #3: Yes

3. Has the statistical analysis been performed appropriately and rigorously?

Reviewer #3: Yes

4. Have the authors made all data underlying the findings in their manuscript fully available (please refer to the Data Availability Statement at the start of the manuscript PDF file)?

Reviewer #3: Yes

5. Is the manuscript presented in an intelligible fashion and written in standard English?

Reviewer #3: Yes

Reviewer #3: I think the standard of the paper is good. The authors need to address all the comments made for the paper to be published.

**Do you want your identity to be public for this peer review?** For information about this choice, including consent withdrawal, please see our Privacy Policy

Reviewer #3: **Yes:** Samuel Kojo Antobam

---

## [Decision Letter · Decision Letter 2]

11 Dec 2025

Geospatial variation and risk factors for malnutrition among postpartum women in rural Bangladesh

PGPH-D-24-03004R2

Dear Dr Bellows,

We are pleased to inform you that your manuscript 'Geospatial variation and risk factors for malnutrition among postpartum women in rural Bangladesh' has been provisionally accepted for publication in PLOS Global Public Health.

Best regards,

Julia Robinson

Executive Editor

Reviewer Comments (if any, and for reference):

Reviewer's Responses to Questions

**Comments to the Author**

Reviewer #3: All comments have been addressed

Reviewer #4: All comments have been addressed

publication criteria?

Reviewer #3: Yes

Reviewer #4: (No Response)

3. Has the statistical analysis been performed appropriately and rigorously?

Reviewer #3: Yes

Reviewer #4: (No Response)

4. Have the authors made all data underlying the findings in their manuscript fully available (please refer to the Data Availability Statement at the start of the manuscript PDF file)?

Reviewer #3: Yes

Reviewer #4: (No Response)

5. Is the manuscript presented in an intelligible fashion and written in standard English?

Reviewer #3: Yes

Reviewer #4: (No Response)

Reviewer #3: I am satisfied with how the authors addressed the comments. They have taken their time to deal with all the issues rasied thoroughly

Reviewer #4: This manuscript is mostly descriptive as is the usual format for this type of investigation. The Statistical Analysis section is adequately detailed with the usual geospatial analysis tools with appropriate indices. This appears to be a good exploration of spatial dependency of nutritional status within the study area.

The paper is generally well written and the statistical input, with revisions, is reasonably applied and, as noted above, is generally well written. Corrected components were the Spatial Analysis Component, Methodological Approach (particularly the use of multinomial logistic regression) , the five-point Likert scale for food security assessment and k-function justification. The tables, figures and supplemental materials were well formatted, foot noted and interpretable.

**Do you want your identity to be public for this peer review?** For information about this choice, including consent withdrawal, please see our Privacy Policy

Reviewer #3: **Yes:** Samuel Kojo Antobam

Reviewer #4: No
